# Pravastatin Improves Colonic and Hepatic Microcirculatory Oxygenation during Sepsis without Affecting Mitochondrial Function and ROS Production in Rats

**DOI:** 10.3390/ijms24065455

**Published:** 2023-03-13

**Authors:** Anne Kuebart, Katharina Gross, Jan-Joschua Ripkens, Theresa Tenge, Annika Raupach, Jan Schulz, Richard Truse, Stefan Hof, Carsten Marcus, Christian Vollmer, Inge Bauer, Olaf Picker, Anna Herminghaus

**Affiliations:** Department of Anesthesiology, University Hospital Duesseldorf, Moorenstrasse 5, 40225 Duesseldorf, Germany

**Keywords:** sepsis, pravastatin, microcirculation, mitochondrial function

## Abstract

Microcirculatory and mitochondrial dysfunction are considered the main mechanisms of septic shock. Studies suggest that statins modulate inflammatory response, microcirculation, and mitochondrial function, possibly through their action on peroxisome proliferator-activated receptor alpha (PPAR-α). The aim of this study was to examine the effects of pravastatin on microcirculation and mitochondrial function in the liver and colon and the role of PPAR-α under septic conditions. This study was performed with the approval of the local animal care and use committee. Forty Wistar rats were randomly divided into 4 groups: sepsis (colon ascendens stent peritonitis, CASP) without treatment as control, sepsis + pravastatin, sepsis + PPAR-α-blocker GW6471, and sepsis + pravastatin + GW6471. Pravastatin (200 µg/kg s.c.) and GW6471 (1 mg/kg) were applied 18 h before CASP-operation. 24 h after initial surgery, a relaparotomy was performed, followed by a 90 min observation period for assessment of microcirculatory oxygenation (μHbO_2_) of the liver and colon. At the end of the experiments, animals were euthanized, and the colon and liver were harvested. Mitochondrial function was measured in tissue homogenates using oximetry. The ADP/O ratio and respiratory control index (RCI) for complexes I and II were calculated. Reactive oxygen species (ROS) production was assessed using the malondialdehyde (MDA)-Assay. Statistics: two-way analysis of variance (ANOVA) + Tukey’s/Dunnett’s post hoc test for microcirculatory data, Kruskal–Wallis test + Dunn’s post hoc test for all other data. In control septic animals µHbO_2_ in liver and colon deteriorated over time (µHbO_2_: −9.8 ± 7.5%* and −7.6 ± 3.3%* vs. baseline, respectively), whereas after pravastatin and pravastatin + GW6471 treatment μHbO_2_ remained constant (liver: µHbO_2_ pravastatin: −4.21 ± 11.7%, pravastatin + GW6471: −0.08 ± 10.3%; colon: µHbO_2_ pravastatin: −0.13 ± 7.6%, pravastatin + GW6471: −3.00 ± 11.24%). In both organs, RCI and ADP/O were similar across all groups. The MDA concentration remained unchanged in all groups. Therefore, we conclude that under septic conditions pravastatin improves microcirculation in the colon and liver, and this seems independent of PPAR-α and without affecting mitochondrial function.

## 1. Introduction

Sepsis remains one of the most challenging conditions to treat in intensive care units. Its high mortality rate of around 25% emphasizes the urgent need for a comprehensive understanding of the pathomechanisms [1].

Sepsis is defined as an inadequate host response to an inflammatory stimulus that causes potentially life-threatening organ dysfunction [2]. The underlying macro- and microcirculatory disturbances result in oxygen deficiency, marked by lactate increase and organ failure [3,4]. Besides other organ injuries, intestinal and liver dysfunction during sepsis impair the patient’s outcome directly. Blaser et al. found a significant correlation between gastrointestinal symptoms and increased 28 d mortality [5]. This is in line with the theory of Haussner et al., describing multiple mechanisms of mucosal barrier dysfunction during sepsis [6]. As the gut hosts the body’s largest microbiome, an oxygen deficiency resulting from impaired microcirculation or dysfunctional mitochondria might cause a breakdown of the intestinal blood barrier and open a portal for the entry of pathogen-associated molecular patterns (PAMPs), emphasizing the importance of maintaining this barrier. Similarly, liver function is impaired during sepsis in 34% to 46% of all patients [7]. Hepatic dysfunction in sepsis is an independent predictor of mortality as it increases the mortality rate to 54% [8,9]. In this case, further microcirculatory and mitochondrial disturbances have been discussed as underlying mechanisms [10].

In the past few years, several drugs routinely used in the intensive care setting have been evaluated regarding their impact on the course of sepsis. Among them, statins have gained greater attention. Statins, such as pravastatin or simvastatin, are primarily used to treat hypercholesteremia, but they also show pleiotropic effects in the state of sepsis. In the past, it was demonstrated that they can have direct anti-inflammatory effects [11]. In line with this, several studies have shown a positive impact of statins on sepsis outcomes, but the level of evidence remains weak [12,13,14,15]. Therefore, this study aims to further elucidate the mechanisms behind statins’ effects, as the exact pathways of statins’ action are still not fully understood. However, the primary mechanism—the inhibition of 3-hydroxy-3-methylglutaryl-coenzyme A (HmG-CoA)-reductase—has led statins to show multiple pleiotropic effects such as facilitating higher nitric oxide (NO) levels as well as anti-oxidant and anti-inflammatory effects [16]. One possible mechanism of these pleiotropic beneficial effects of statins may be the regulation of the peroxisome proliferator-activated receptor alpha (PPAR-α) [17]. The PPAR-α receptor belongs to the nuclear receptor superfamily and is expressed in the liver, heart, brain, kidney, testis, and intestine [18,19]. PPAR-α regulates several hepatocyte genes, especially those involved in mitochondrial metabolic functions such as fatty acid oxidation [20]. Statins have been shown to increase mRNA levels and the activation of PPAR-α in liver cells. Moreover, statins can positively modulate vasomotor function by limiting acetylcholine-induced vasoconstriction [21,22] as well as improving peripheral NO-mediated vascular relaxation [23]. Nagaoka et al. revealed another positive effect of statins on microcirculation: an increase in blood velocity and flow in the retinal arterioles [24]. Whether these effects occur in other organs and under septic conditions has not been examined so far and is addressed in the current study. Furthermore, the statin-induced alteration in mitochondrial function is also not fully understood and is even controversial. Their effects are described as both positive [25] and negative [26] and differ depending on the tissue, species, and study design. Therefore, the current study aims to evaluate the effects of statins on intestinal and hepatic microcirculation and mitochondrial function and their possible role in PPAR-α-blockade in these interactions.

Using a well-established septic animal model, the effect of pravastatin alone and in combination with a PPAR-α inhibitor on the microcirculation was assessed. Further, drug-induced alterations in mitochondrial function and ROS levels were monitored. This offers insights into new therapeutic approaches targeting microcirculation, ROS production, and dysfunctional mitochondria in septic patients [27].

## 2. Results

### 2.1. Vital Parameters/Organ Damage Parameters

Haemodynamic variables: mean arterial pressure and heart rate were not significantly different between the groups of septic animals at any time point, but they differed in the time course (Table 1). In the control group, after 60 min mean arterial blood pressure (MAP) was lower compared to baseline. Further, in the pravastatin group, the heart rate (HR) was decreased after 30, 60, and 90 min compared to the baseline, and after 90 min HR was also decreased in the control and pravastatin/GW6471 groups.

Lactate concentration was measured via blood gas analyses every 30 min and compared between the groups. No significant differences were observed between the groups, but the values differed over time. In all groups, the lactate concentration decreased after 60 and 90 min, and in the Pravastatin/GW6471 group also after 30 min compared to the baseline (Table 1).

Organ damage parameters such as creatinine, urea, alanine aminotransferase (ALT), aspartate aminotransferase (AST), and lactate dehydrogenase (LDH) were determined. No significant differences were observed between the groups (Table 2).

### 2.2. Microcirculation of the Colon

In control septic animals, µHbO_2_ decreased 60 min and 90 min after laparotomy, compared to the baseline. Pravastatin with or without combination with PPAR-α antagonist GW6471 could prevent this decrease in µHbO_2_. After treatment with GW6471 alone, a significant decrease in µHbO_2_ was observed, similar to that observed in untreated septic animals (Figure 1).

### 2.3. Microcirculation of the Liver

Similar to the colon, the liver parenchyma of untreated septic animals showed a decrease in µHbO_2_ over time. Notably, the decrease in µHbO_2_ in the liver was observed about 30 min later than in the colon of septic animals. Here, the decrease was seen in the untreated septic animals and the animals treated with GW6471 90 min after laparotomy. The animals treated with pravastatin or the combination of pravastatin + GW6471 did not show a significant decrease of µHbO_2_ (Figure 2).

### 2.4. Mitochondrial Respiration in the Colon

In the colon, there were no visible alterations concerning mitochondrial function after stimulation of the respiratory chain through complex I and II. RCI and ADP/O stayed unchanged compared to the control group and between the groups (Figure 3A–D).

### 2.5. Mitochondrial Respiration in the Liver

Pravastatin and GW6471 did not change mitochondrial function in the liver significantly. Neither RCI, nor ADP/O were affected by these drugs (Figure 4A–D).

### 2.6. ATP Concentration and Oxidative Stress in the Colon and in the Liver

In order to further elucidate whether pravastatin affects ATP synthesis and oxidative stress, measurements of ATP concentration and MDA assays were conducted. There were no differences between the groups regarding ATP content in the colon and in the liver (Figure 5A,B).

MDA levels in both organs stayed unchanged in all groups compared to the control group as well as between the groups (Figure 5C,D).

## 3. Discussion

In the past, several clinical and experimental studies reported a protective effect of statins in the state of sepsis, even if the level of evidence remains weak [12,13,28,29]. However, the exact mechanisms leading to the observed protective effects are still not fully understood. The results regarding the impact of statins on mitochondrial function have not been fully elucidated either and are even now controversially discussed. The conducted study aimed to reveal whether the positive effects of statins are mediated by influencing microcirculation and mitochondrial function, and further, to examine if a dependency on PPAR-α is apparent. Therefore, septic animals (CASP model) were treated with pravastatin, PPAR-α inhibitor GW6471, or a combination of both.

The main results can be highlighted as follows:Colonic and hepatic microcirculatory variables are significantly decreased under untreated septic conditions, at least after 60 min until the end of the observation period.Pre-treatment with pravastatin prevents microcirculatory impairment and maintains tissue oxygenation in the colon and liver.This observed protective effect of pravastatin is not affected by the PPAR-α-inhibitor GW6471.Neither pravastatin, nor the blocking of its presumed receptor, nor the combination of both, affects colonic or hepatic mitochondrial function. The levels of oxidative stress in the colon and liver do not differ significantly between the intervention groups either.

In this study, the effects of pravastatin and a PPAR-α antagonist during sepsis were studied on Wistar rats undergoing a CASP operation. The main goal was to evaluate changes in microcirculation and mitochondrial function in two organs: the liver and colon, in response to treatment with pravastatin and a PPAR-α antagonist.

For studying these pravastatin mediated effects, a dosage of pravastatin (200 µg/kg b.w., s.c.) in the range of clinically used dosages was chosen based on the literature [28,29,30]. The clinically used daily dosages of 10 mg up to 80 mg/d have proven lipid lowering ability [31] and reflect doses of ~150–1200 µg/kg (orally administered) with an oral bioavailability of ~20% [32]. The dosage of PPAR-α antagonist GW6471 (1 mg/kg) was also chosen based on the literature [18,33,34,35,36].

For this study, the CASP model was chosen as it is still one of the preferred models for studying polymicrobial abdominal sepsis pathophysiology [37,38]. Here, a moderate, sublethal sepsis was induced by the insertion of two 16G stents in the colon ascendens of rats. 24 h after CASP surgery, we observed clearly visible signs of peritonitis. Further, prior analyses already showed elevated cytokine levels of tumor necrosis factor alpha, interleukin 6 and 10 in the here used standardized CASP model [39,40]. This is in line with the literature, showing that a stent of 16G results in a significant increase of cytokines 12 h after CASP surgery, with a mortality rate of 71% 48 h after CASP surgery but stable vital parameters until 1 h before death [41]. In previous studies, CASP-operated animals did not show significant differences in heart frequency, MAP, and metabolic variables at 24 h after induction of sepsis compared to sham animals [39,42]. In this study, the macrocirculatory variables such as heart rate, arterial pressure, and lactate levels remained stable during the experiment. Actually, lactate concentrations showed significant changes in all groups, but all values remained within the physiological range and therefore are not considered to be of clinical relevance. The low mortality rate together with stable vital parameters make this model suitable for measurements during prolonged interventions as performed here [41] and avoid biased results of impaired microcirculation resulting from macrocirculatory disturbances. Sham animals of the CASP model were evaluated in previous analyses and did not differ significantly in vital parameters, cytokine levels, or microcirculatory variables in the course of the experiment compared to the baseline levels [39,42,43]. However, the missing sham data has to be mentioned as a limitation here, as it only allows the interpretation of the effects of pravastatin in the inflammatory context but not in the healthy state.

In this study, we showed a reproducible time-dependent decrease of µHbO_2_ in the colon and liver parenchyma of CASP-operated animals but no signs of organ damage. Previous studies of our group revealed that this decrease is limited to the CASP group, whereas sham animals did not show any significant changes in microcirculatory variables [39,42]. Regarding liver parenchyma, the observed µHbO_2_ decrease of about 10% did not lead to increased enzyme levels (AST/ALT), which serve as liver damage parameters. Thus, we could not observe signs of liver cell damage. Intestinal damage is more difficult to access, as no routinely used clinical biomarkers of intestinal failure are available. Regarding intestinal failure, no additional parameters such as D-lactate, citrulline, or tight-junction parameters have been the subject of this analysis; therefore, the level of intestinal damage induced by the observed decrease in intestinal µHbO_2_ remains speculative. This leads to the question of the relevance of the compromised microcirculation seen here, lacking direct organ damage correlation. Notably, impaired microcirculation itself remains an independent indicator for a worse outcome in sepsis [44], and therapeutic approaches preventing or minimizing its occurrence might present a game-changing tool that therefore has to be investigated carefully. Further, the time point of microcirculatory changes suspected in abdominal sepsis seems to be clinically relevant. Tascon et al. indicate that changes in microcirculatory variables are only reversible if they are detected early [45]. Finally, Dubin et al. reported that fluid resuscitation did restore sublingual and serosal microcirculation but failed to improve mucosal microcirculation, resulting in remaining hypoperfused villi and opening the door for intestinal barrier dysfunction and aggravated sepsis progression [46]. Taken together, the beneficial impact of pravastatin on the microcirculation might have clinical relevance but needs further investigation in clinical studies. Interestingly, the application of the PPAR-α inhibitor did not impair the positive effect of pravastatin on colonic and hepatic microcirculation. Of note, when PPAR-α inhibitor (GW6471) was administered alone, a significant decrease in µHbO_2_ was detected, similar to untreated septic animals, indicating a PPAR-α-independent protective effect of pravastatin. As other pathways have to be evaluated to clarify this observed phenomenon, a possible mode of action could be through the regulation of the endothelial nitric oxide synthase (eNOS) pathway. Most recently, Ren et al. observed a modulation of the Caveolin-1/eNOS pathway by pravastatin, resulting in higher expression of tight junctions and clinically ameliorating acute lung injury during sepsis by decreasing pulmonary microvascular permeability [15]. The regulation of nitric oxide signaling by statins is already a subject of discussion and was recently reviewed [47].

Han et al. provide another indirect evidence for a PPAR-α independent mode of action after pravastatin administration under septic conditions [48]. They discovered a downregulation of PPAR-α in cardiomyocytes after sepsis induction. If this reduction of receptor density occurred not only in cardiomyocytes but also in hepatocytes and enterocytes, it would be another possible indication that the protective effects induced by pravastatin are not mediated by PPAR-α.

In our study, pravastatin affected neither hepatic nor colonic mitochondrial function. In our previous in vitro study, we could show that this drug has a deteriorating effect on hepatic mitochondria but a rather positive influence on colonic mitochondrial respiration [49]. These different results could be explained by divergent study designs (in vitro vs. in vivo), varying conditions (septic animals vs. tissues from healthy rats), and possibly unequal drug concentrations (fully controlled target drug concentration in vitro vs. possibly variable drug concentration limited by macro- and microcirculatory alterations under septic conditions). Regarding the lack of effects of pravastatin on mitochondrial function, it is not possible to speculate about the role of PPAR-α in this context.

As the respiratory chain is the main source of ROS in cells, the similar MDA levels in all groups seem to be a logical consequence of unchanged mitochondrial function in treated and untreated animals. Of note, mitochondrial ROS production differs strongly between tissues according to their metabolic activity and could have a positive effect (mitohormesis) or deteriorating consequences, including cell death [50].

In conclusion, the positive effect of pravastatin on microcirculation seems to be independent of PPAR-α activation. As the results of mitochondrial respiration measurements and oxidative stress level analyses in the liver and in the colon do not differ in treated and untreated septic animals, the protective effect of pravastatin does not seem to be conveyed by changes in mitochondrial respiration. Further studies are necessary to evaluate the possible mechanisms of statins’ action on the hepatic and colonic microcirculation as well as on the mitochondrial function under septic conditions.

## 4. Material and Methods

### 4.1. Animal Experiments

Male Wistar rats (320–380 g body weight (b.w.), n = 40) were used. All experiments were conducted after approval by the local animal care and use committee (North Rhine-Westphalia Office of Nature, Environment, and Consumer Protection, Recklinghausen, Germany, AZ. 84-02.04.2015.A398) and in accordance with the ARRIVE guidelines. Animals were kept in standardized conditions (day/night-cycle of 12 h, 20–22 °C) and fed with standardized food pellets (Ssniff Spezialdiäten GmbH, Soest, Germany) and water ad libitum. Animals were randomly assigned to one of the four experimental groups: (1) control: vehicle (natrium chloride (NaCl)/dimethyl sulfoxide (DMSO); (2) pravastatin; (3) GW6471; and (4) pravastatin + GW6471. Pravastatin (200 µg/kg b.w.) or vehicle solution was injected s.c. 18 h prior to CASP-operation. To determine the role of the PPAR-α, PPAR-α-antagonist GW6471 was injected i.p. (1 mg/kg b.w., in 5% DMSO) into a subgroup of animals 30 min before pravastatin application.

In order to induce abdominal sepsis, the colon ascendens stent peritonitis (CASP) model was used as described before [39,40]. In addition, animals were anaesthetized using sevoflurane per inhalation, and buprenorphine (0.05 mg/kg b.w.,s.c.) was applied for analgesia. The successful induction of anaesthesia was confirmed by the absence of reaction reflexes to a standardized stimulus (toe pinch) and by the absence of any movement during the intervention. Rats were placed on a warming mat, and dexpanthenol eye ointment was applied to ensure eye protection. Under sterile conditions, a 2 cm median laparotomy was performed. The caecum was mobilized, and two 16G plastic stents were placed on the antimesenteric side (ileocaecal transition and 2 cm cranial) and anchored via single knots. After repositioning the caecum, the abdomen was closed, and 5 mL 0.9% NaCl solution was administered. Animals received buprenorphine (0.05 mg/kg b.w.) and volume administration s.c. and were scored using the predefined Septic Rat Severity Score (SRSS). The SRSS is based on the Murine Sepsis Score [51] and was adjusted for rats in several previous studies by our working group [40,52,53]. Animals reaching the maximal allowed score of 10 points were euthanized by administering sodium pentobarbital (120 mg/kg b.w., i.p).

Twenty-four hours after sepsis induction via CASP-operation, haemodynamic measurements and assessment of microcirculation were conducted for two hours (30 min for stabilization and 90 min experiment). Rats were again anaesthetized with sodium pentobarbital (60 mg/kg b.w., i.p.) and buprenorphine (0.05 mg/kg b.w., s.c.) injections. A heat mat maintained the animals’ normothermia, and vital parameters were evaluated by invasive blood pressure measurement and volume-controlled ventilation after a tracheotomy. Anaesthesia was maintained by continuous pentobarbital infusion (10 mg/kg b.w./h) via a central venous catheter. After reaching an adequate depth of anaesthesia, a re-laparotomy was performed. Pancuronium bromide (3 mg/kg b.w.) suppressed spontaneous breathing. For the point-of-care diagnostic, blood (90 µL) was withdrawn from an arterial catheter every 30 min for blood gas analysis (ABL 800 flex, Radiometer, Copenhagen, Denmark). After 90 min of measurements, the animals were euthanized under deep anaesthesia by exsanguination, and the liver and colon were removed. Tissue samples were harvested for measurement of mitochondrial function and placed immediately in a cold isolation puffer (200 mM mannitol, 50 mM sucrose; 5 mM KH_2_PO_2_; 5 mM 3-(N-morpholino)-propanesulfonic acid (3-MODS); 0.1% bovine serum albumin (BSA), S; 1mM ethylene glykol-bis-(beta-aminoethylether)-tetraacetic acid (EGTA), pH 7.15), as well as flash frozen in liquid nitrogen and further stored at −80 °C for later assessment of malondialdehyde (MDA) and ATP concentrations. Blood samples were analyzed for the levels of organ damage parameters such as creatinine, urea, aspartate aminotransferase (AST), and alanine aminotransferase (ALT). The sequence of the experimental procedures is presented in Figure 6.

### 4.2. Evaluation of Microcirculatory Variables

Microcirculatory oxygenation (µHbO_2_) of the colon and liver was evaluated by light-spectrophotometry (O2C LW 2222, Lea Medizintechnik GmbH, Gießen, Germany) as has been described before [28,39]. By placing an O2C sensor directly on the parenchyma, the postcapillary oxygenation µHbO_2_ was captured in percent through the tissue with a depth of 0.7 mm. These values were analyzed every two seconds and summarized as means for periods of 5 min. The µHbO_2_ values reported are the means of the last 5 min every 30 min. Further, µHbO_2_ measures are displayed as ∆µHbO_2,_ calculated as µHbO_2_ (baseline)-µHbO_2_ (0, 30, 60, and 90 min).

Additionally, white light (450–1000 nm) and laser light (820 nm, 30 mW) were transmitted to the tissues via a microlightguide, and the reflected light was analyzed. The wavelength-dependent absorption of the applied light is used to calculate the percentage of oxygenated hemoglobin in the microcirculation (µHbO_2_). As the biggest fraction of the blood volume is stored in venous vessels (85%), postcapillary oxygenation represents the critical partial pressure of oxygen for hypoxia [54].

### 4.3. Tissue Homogenization

The extracted colon was immediately placed in an ice-cold isolation buffer, cleaned from feces and mucus, and incubated with trypsin 0.05% for 5 min. The tissue was transferred into an isolation buffer consisting of 2% bovine serum albumin (BSA) and protease inhibitors (cOmplete™ Protease Inhibitor Cocktail, Roche Life Science, Mannheim, Germany). The tissue was further cut into small pieces and homogenized (Potter-Elvehjem, 5×, 2000 rpm). Liver tissue was treated similarly. Extracted liver tissue was subsequently placed in an isolation buffer and shredded into 2–3 mm^3^ pieces. After replacing the buffer, the tissue was homogenized (Potter-Elvehjem, 5×, 2000 rpm). The protein content of both tissue homogenates was determined using the Lowry standard [55]. All steps were performed on ice using buffers at 4 °C.

### 4.4. Determination of Mitochondrial Respiratory Rates

Mitochondrial respiratory rate was determined as described previously [40,56,57]. In addition, the measurements were done using a Clark-type electrode (model 782, Strathkelvin Instruments, Glasgow, Scotland). The tissue homogenates were used at a protein concentration of 4 mg/mL or 6 mg/mL for the liver and colon, respectively. These concentrations were obtained by adding respiratory medium (130 mM KCl, 5 mM K_2_HPO_4_, 20 mM MOPS, 2.5 mM EGTA, 1 µM Na_4_P_2_O_7_, 0.1% BSA for liver, and 2% BSA for colon, pH 7.15) to the tissue homogenates. All samples were examined at a constant temperature (30 °C), ensuring a constant oxygen solubility (223 µmol O_2_/L, as stated in the manufacturer protocol).

The mitochondrial state 2 respiratory rate was evaluated by adding either the complex I substrates glutamate (Fluka, München, Germany) and malate (Serva Electrophoresis GmbH, Heidelberg, Germany) (both 2.5 mM) or the complex II substrate succinate (Sigma-Aldrich Corporation, St. Louis, MO, USA) (10 mM for the liver, 5 mM for the colon) after adding 0.5 µM rotenone (Sigma-Aldrich Corporation, St. Louis, MO, USA), which serves as an inhibitor of complex I activity.

Mitochondrial state 3 respiratory rate, defined as the maximal coupled mitochondrial respiration, was measured after adding adenosine diphosphate, 250 μM for the liver and 50 μM for the colon (ADP, Sigma-Aldrich Corporation, St. Louis, MO, USA).

The results of state 2, reflecting the electron transport system (ETS), were related to the state 3 results, which indicate oxidative phosphorylation (OXPHOS) by calculating the respiratory control index (RCI, state 3/state 2) as an indicator for the coupling of ETS and OXPHOS. Further, using the added ADP amount in state 3 and the consumed oxygen, the OXPHOS-efficacy as an ADP/O ratio was determined.

### 4.5. Malondialdehyde-Assay

As a surrogate for lipid peroxidation/oxidative stress, malondialdehyde (MDA) was quantified using the thiobarbituric (TBA) acid assay. Liver and colon tissue (approximately 50 mg) stored at −80 °C, was thawed and homogenized in 500 µL 1.15% potassium chloride. Further, 250 µL of tissue homogenate was mixed with 1.5 mL of 1% phosphoric acid and 0.5 mL of 0.6% TBA. Samples were incubated at 95 °C for 45 min and stored on ice afterwards. After adding 2 mL of butanol, the samples were centrifuged (for 15 min 2900× *g*). Again, 200 µL sample volume was taken and added to an equal volume of potassium chloride. Absorbance was analyzed by photospectrometry at a wavelength of 535 nm and 520 nm. The MDA concentration was calculated using an MDA-standard and normalized to protein concentration determined by the Lowry method and expressed as nanomole MDA per milligram protein.

### 4.6. ATP-Measurement

Tissue samples frozen in liquid nitrogen (approx. 50 mg) were immediately homogenized with a 3-fold volume (µL) of Tris-HCl buffer (20 mM Tris, 135 mM KCl, pH 7.4). 450 µL of boiling 100 mM Tris/4 mM ethylenediaminetetraacetic acid (EDTA) buffer (pH 7.75) were added to 50 µL homogenate, incubated for 2 min at 100 °C, and centrifuged at 1000× *g* for 2 min. ATP was determined by the ATP Bioluminescence Assay Kit CLS II (Roche, Basel, Switzerland) using luciferase reagent. The data are presented as nanomoles of ATP per milligram of protein.

### 4.7. Plasma Analyses

The plasma was obtained by centrifugation (4 °C, 4000× *g*, 10 min) of blood samples collected in EDTA tubes and stored at −80 °C. Activities of alanine aminotransferase (ALT), aspartate aminotransferase (AST), and lactate dehydrogenase (LDH), as well as concentrations of creatinine and urea, were measured in the Central Institute of Clinical Chemistry and Laboratory Medicine of the University Hospital Duesseldorf, Germany.

### 4.8. Statistics

In order to calculate the appropriate sample size, an a priori power analysis (G*Power Version 3.1.7, Heinrich Heine University, Düsseldorf, Germany) was performed. With n = 10 animals per group at a given α ≤ 0.05 (two-tailed) and an expected mean difference in μHbO_2_ of at least 20% (percentage points) with an expected standard deviation of 10–15% (based on previous studies), a power of 84.5% resulted.

GraphPad Prism v6.01 (GraphPad Software, Inc., Boston, MA, USA) was used for further statistical analysis. Microcirculatory data were analyzed with a two-way analysis of variance (ANOVA) for repeated measures, followed by Dunnett’s post-hoc test for differences versus baseline and Tukey’s post-hoc test for differences between groups. Wherever delta values are presented, the absolute baseline value was subtracted from the absolute value at the respective observation points to individualize the data to each rat’s baseline. Therefore, the data are provided as absolute percentage points with regard to baseline values and not as relative changes. All other data were analyzed with the Kruskal-Wallis-Test followed by Dunn’s post hoc test after failing the Kolmogorov-Smirnov-Test for a normal distribution. The data are shown as mean ± SD (for the microcirculatory data) and as individual values with medians and interquartile ranges (for all the other data). *p* < 0.05 was considered significant.

## 5. Limitations

The experiments described above were conducted in rats, and so care should be taken in interpreting the obtained results in a clinical context. Further, as stated above, the missing sham data has to be mentioned as a limitation and allows the interpretation of pravastatin effects only in septic conditions, not in a healthy state. Another limitation concerns the here used PPAR-α inhibitor GW6471, which is in fact widely used at the prescribed dosage. Nevertheless, to our best knowledge, no study has examined the successful PPAR-α inhibition by reduced transcription levels of its target proteins so far. As a proven inhibition of PPAR-α enables a clear conclusion regarding the dependency or independency of the observed pravastatin-mediated effects, this should be the subject of further studies [18,33,34,35,36].

## Figures and Tables

**Figure 1 ijms-24-05455-f001:**
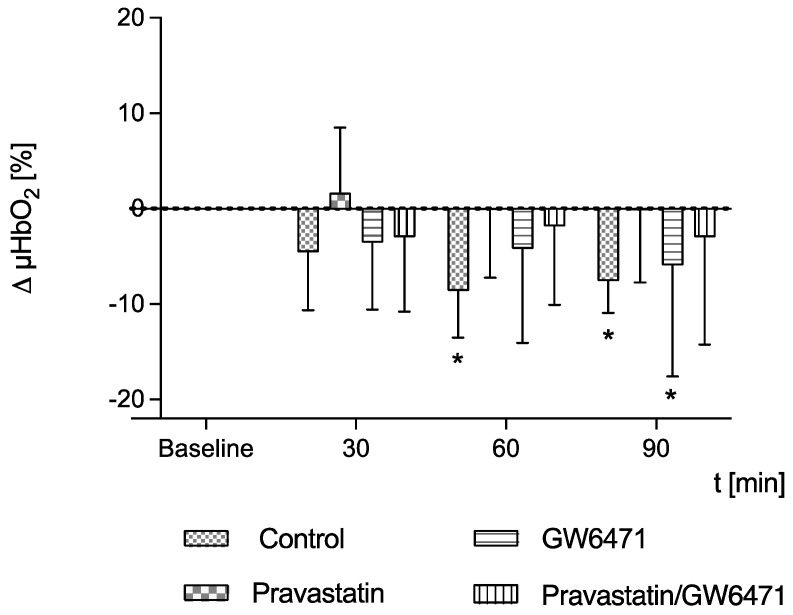
Postcapillary oxygenation of the colon (µHbO_2_) was measured in untreated septic animals (control) and after treatment with pravastatin, PPAR-α-antagonist GW6471, pravastatin and GW6471. Data are presented as absolute percentage points with regard to baseline values and shown as mean + SD, n = 10, * *p* < 0.05 vs. baseline.

**Figure 2 ijms-24-05455-f002:**
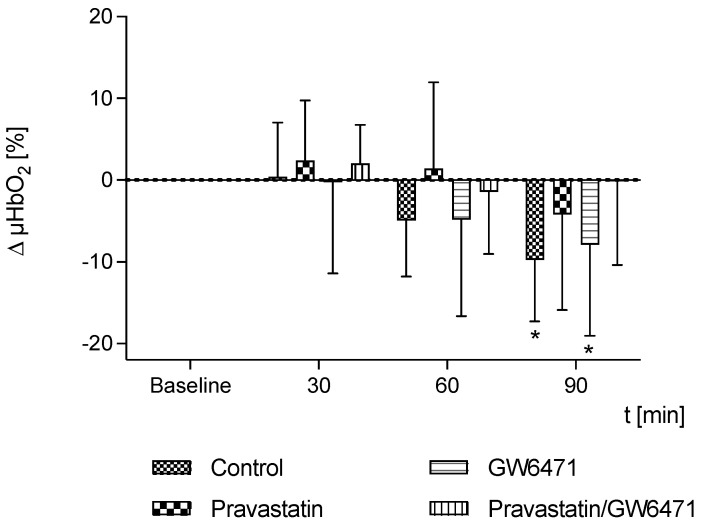
Postcapillary oxygenation of the liver (µHbO_2_) was measured in untreated septic animals (control) and after treatment with pravastatin, PPAR-α-antagonist GW6471, pravastatin and GW6471. Data are presented as absolute percentage points with regard to baseline values and shown as mean + SD, n = 10, * *p* < 0.05 vs. baseline.

**Figure 3 ijms-24-05455-f003:**
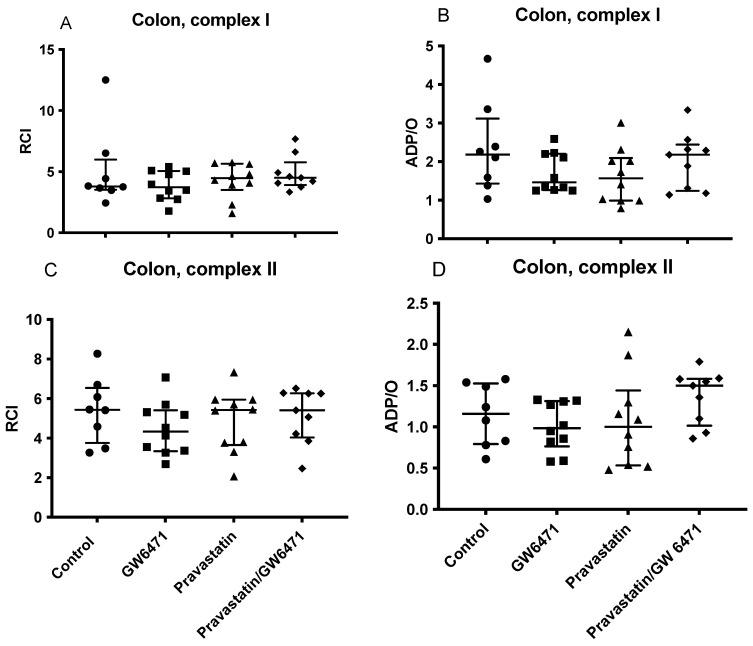
Effect of pravastatin, GW6471, and the combination of both on colonic mitochondrial respiration: RCI (state 3/state 2) (**A**,**C**), (ADP/O = ADP added/oxygen consumed in state 3) (**B**,**D**). All mitochondrial parameters were analysed for complex I (**A**,**B**) and II (**C**,**D**). Data are presented as individual values, median with interquartile range n = 8–10.

**Figure 4 ijms-24-05455-f004:**
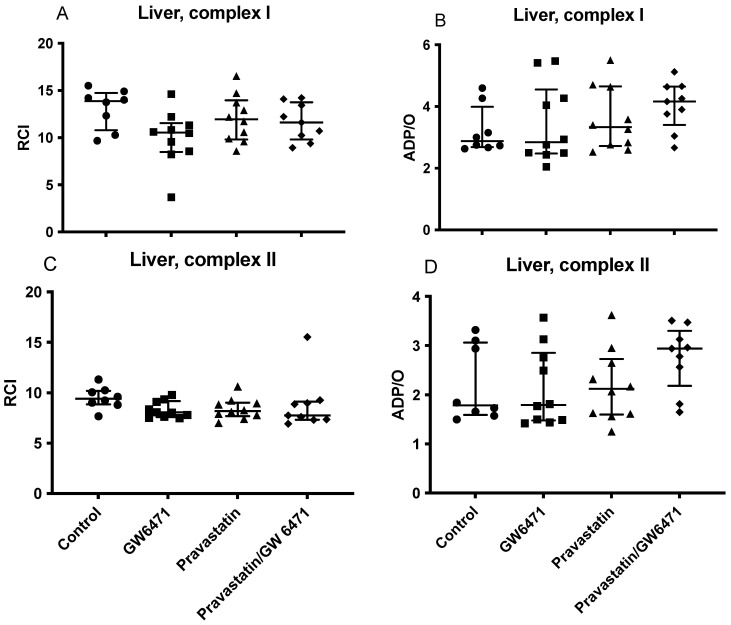
Effect of pravastatin, GW6471, and the combination of both on hepatic mitochondrial respiration: RCI (state 3/state 2) (**A**,**C**), (ADP/O = ADP added/oxygen consumed in state 3) (**B**,**D**). All mitochondrial parameters were analysed for complex I (**A**,**B**) and II (**C**,**D**). Data are presented as individual values, median with interquartile range n = 8–10.

**Figure 5 ijms-24-05455-f005:**
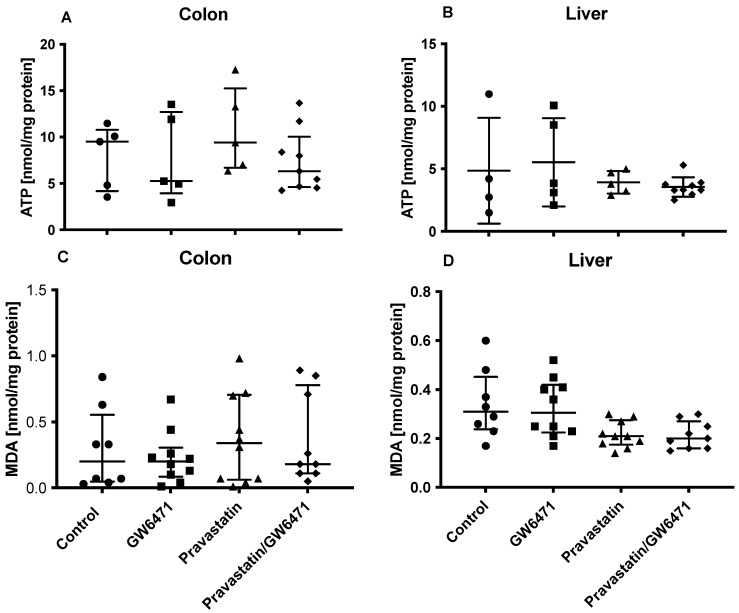
ATP concentration (**A**,**B**) and malondialdehyde (MDA) level (**C**,**D**) in colon and liver tissue. Animals were treated with Pravastatin, GW6471 or a combination of both substances prior to colon ascendens stent peritonitis (CASP) operation. ATP concentration was determined by ATP Bioluminescence Assay kit CLS II (n = 4–9). MDA concentrations (nmol/mg protein) were assessed as described before (n = 8–10). The data are presented as individual values with a median and interquartile range.

**Figure 6 ijms-24-05455-f006:**
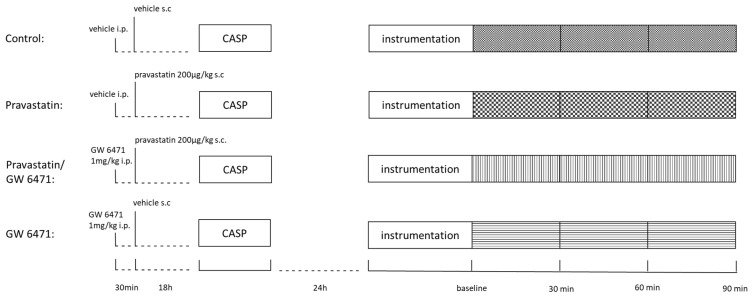
The sequence of experimental procedures. Pravastatin or vehicle were injected s.c. 18 h before the colon ascendens stent peritonitis (CASP)-operation, the PPAR-α-blocker GW6471 30 min prior the application of pravastatin. The experiments were performed 24 h after the CASP-operation.

**Table 1 ijms-24-05455-t001:** Mean arterial blood pressure (MAP), heart rate (HR) and lactate concentration in control septic animals (control) and after treatment with pravastatin (Pravastatin), PPAR-α-antagonist GW6471 (GW6471) and a combination with both drugs (Pravastatin + GW6471). n = 10, * *p* < 0.05 vs. baseline, ^§^
*p* < 0.05 vs. 30 min, ^&^
*p* < 0.05 vs. 60 min.

Mean Arterial Pressure (MAP) [mmHg]
	Control	Pravastatin	GW6471	Pravastatin + GW6471
Baseline	110.44 ± 23.27	98.94 ± 29.30	102.73 ± 30.77	103.93 ± 32.21
30 min	101.06 ± 23.05	93.59 ± 34.26	100.26 ± 26.29	108.82 ± 33.93
60 min	93.13 ± 37.30 *	98.09 ± 40.95	90.04 ± 30.90	104.66 ± 37.12
90 min	99.27 ± 38.59	98.40 ± 45.79	110.27 ± 36.32 ^&^	107.75 ± 37.81
Heart rate (HR) [beats/min]
Baseline	469.37 ± 56.21	457.27 ± 32.78	463.62 ± 50.58	468.31 ± 60.39
30 min	437.36 ± 45.91	420.18 ± 58.32 *	458.73 ± 66.80	449.32 ± 58.49
60 min	445.73 ± 95.89	406.42 ± 50.81 *	435.66 ± 66.82	440.30 ± 59.19
90 min	424.94 ± 74.79 *	391.18 ± 63.07 *	453.05 ± 69.41	423.35 ± 70.22 *
Lactate [nmol/L]
Baseline	1.23 ± 0.31	1.51 ± 0.88	1.24 ± 0.41	1.55 ± 0.42
30 min	1.15 ± 0.33	1.29 ± 0.62	1.10 ± 0.42	1.15 ± 0.28 *
60 min	0.96 ± 0.27 *	1.09 ± 0.51 *	0.9 ± 0.29 *	1.13 ± 0.33 *
90 min	0.83 ± 0.29 *^,§^	0.97 ± 0.64 *^,§^	0.75 ± 0.22 *^,§^	0.89 ± 0.21 *^,§^

**Table 2 ijms-24-05455-t002:** Creatinine, urea, aspartate transferase (AST), alanine transferase (ALT) and lactate dehydrogenase (LDH) in control septic animals (control) and after treatment with pravastatin (Pravastatin), PPAR-α-antagonist GW6471 (GW6471) and a combination with both drugs (Pravastatin + GW6471). n = 7–10.

	Control	Pravastatin	GW6471	Pravastatin + GW6471
Creatinine [mg/dL]	0.29 ± 0.06	0.37 ± 0.24	0.27 ± 0.10	0.31 ± 0.09
Urea [mg/dL]	39.43 ± 5.87	42.42 ± 11.79	40.10 ± 8.35	46.81 ± 9.02
AST [U/L]	118.5 ± 102.53	222.70 ± 226.60	84.30 ± 25.50	106.5 ± 48.32
ALT [U/L]	52.5 ± 51.81	118.80 ± 173.70	39.30 ± 20.87	41.60 ± 12.55
LDH [U/L]	335.25 ± 234.45	315.57 ± 231.75	212.56 ± 54.55	235.38 ± 130.04

## Data Availability

The data presented in this study are available on request from the corresponding author.

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
