# Peer review of "Pravastatin Improves Colonic and Hepatic Microcirculatory Oxygenation during Sepsis without Affecting Mitochondrial Function and ROS Production in Rats"

_ijms, 2023, doi:10.3390/ijms24065455_

Round 1

Reviewer 1 Report

Kuebart et al. report improved microcirculatory function in a rat model of moderate abdominal sepsis through pravastatin in a manner independent of PPAR-alpha as judged by the use of a PPAR-alpha inhibitor (GW6471). This lab has an extensive track record in the use of the rat CASP model and most methods applied here. Yet in this report, essential information is missing to regard the evidence sufficient to support the conclusion. Above all, I am concerned about a lack of positive controls in three instances, unless there is more convincing literature to support that such controls can be skipped.

(1) To what extent is the microcirculation in the intestine and liver impaired in this model of moderate sepsis compared to sham animals on the one hand and to severe sepsis or septic shock on the other? Does it lead to organ dysfunction and death, or is it reversible and all animals recover fast? We need to have an idea of this to judge whether the positive effect through the statin is of relevance for the clinical outcome. If microcirculatory dysfunction is minor and short-lived, the statin-effect described may be of little clinical interest yet. It would be very good to have a clinically relevant endpoint, e.g. time to recovery evaluated by the Septic Rat Severity Score. 

(2) Is the statin dose used high enough to lower blood cholesterol in this model, either with standard or high cholesterol diet? That is, is the dose used sufficient to induce the primary effect of statins? Or is it even higher? In the latter case, could it be achieved in humans, i.e., could it be clinically relevant in the first place?

(3) Most importantly, GW6471 does not show a statistically significant effect in this study. But do we know whether it actually inhibited PPAR-alpha in intestine and liver tissues? Are we sure the dose taken from a study on brain capillaries (ref. # 39) inhibits the transcriptional activity of PPAR-alpha here as well? This should be tested by RT-PCR/western blotting for selected targets of the transcription factor.

Further major comments:

(4) Lines 63-66: The authors cite a recent meta-analysis (ref. # 15) that criticizes a weak level of evidence for clinical benefit for statins in infections and go on to claim that "this uncertainty often results from insufficiently understood mechanisms of action". I strongly disagree with this claim. "Clinical evidence" and "knowing the mechanism of action" must be viewed separately: Many experimental drugs have a very well understood mechanism of action but never made it to clinical application due to insufficient clinical evidence for a benefit. And vice versa, there can very well be a high level of clinical evidence supporting the use of a drug without understanding of the mechanism of action. Please revise.

(5) On the same note: At the beginning of the Discussion (line 186) it is mentioned that the protective influence of statins in sepsis is already proven: Please contrast this statement with the meta-analysis in the Introduction.

(6) Line 287: I tried to find the original publication for development and validation of the Septic Rat Severity Score, which is reminiscent of the Murine Sepsis Score by Shrum et al. BMC Res Notes (2014) 7:233. It is not the given self-citation ref. # 27 and none of the references given therein. Please correct. Also, spell out the maximal allowed score.

(7) Lines 71-73: A review is cited (ref. 18) to support the claim that PPAR-alpha is mainly expressed in liver... and intestine. When I checked this review, it listed only the liver among the tissues mainly expressing PPAR-alpha and said that it was also expressed in the intestine among others. However, instead of primary reports, ref. # 18 also only cites an older review by the same authors. The chains of self-citation are annoying in my view, and I request the authors to insert original primary articles to support expression of PPAR-alpha protein in intestine and liver, no matter whether levels can be compared or not. Possibly consider a database such as the human protein atlas (https://www.proteinatlas.org/).

(8) Line 279: Explain "standardized pain stimulus". Or can it be found in the requested reference for the score development/validation (see point 7)?

(9) Title: Please indicate the rat model in the title to this paper to avoid mistaking it for a human study.

(10) I suggest to include a sketch illustrating the sequence of experimental procedures and measurements indicating the point of the baseline measurements.

(11) 4.8 Statistics: It should be mentioned whether the test for normal distribution of experimental data was always passed or sometimes failed. In case it is not passed, a non-parametric test should be used such as the Kruskal-Wallis test with e.g. Dunn's post hoc test (the authors have used this test before).

I am also not sure about another requirement for using two-way ANOVA: homogeneity of variance. Especially in Figure 7, the relative variance without the statin is higher than with statin. The problem is that the bar charts hide any information about data distribution. Since all experimental conditions are clearly labelled in all bar charts, I recommend to not use fill patterns but instead overlay the bars with scatter plots of the data. The data summarized in Table 2 should also be shown in this way (one panel for each parameter). It is unclear why they are summarized in a table.

Minor comments:

(12) Line 60: The term "pluripotent" refers to a property of stem cells. here, the term "pleiotropic", as used later, is already appropriate.

(13) Lines 77-80: A report on the positive effect of statins on retinal microcirculation is mentioned (ref. # 23), and then it is asked whether this effect occurs under septic conditions. It reads as if we expect statins to improve retinal microcirculation in sepsis, not intestine and liver.

Reviewer 2 Report

This study made a sepsis model of abdominal infection by using CASP in rats, and observed the hemodynamic changes, arterial blood lactate levels, liver and kidney function, microcirculation perfusion, tissue oxygenation, mitochondrial respiration and ATP content 24h after mold making. The author give us the following main conclusions: The Pre-treatment with pravastatin prevents microcirculatory impairment and maintain tissue oxygenation in the colon and liver, and this protective effect is not affected by PPAR-α-inhibitor GW6471. The results have certain guiding significance for the prevention and treatment of sepsis.

However, there are several following issues that need to be further explored.

1.This study was limited to controlled studies in the rat sepsis model, and might be better if tested by cell experiments;

2.The animal model lacks the sham operation group, which is slightly insufficient in the control analysis of some basic parameters;

3.The observation starts 24 hours after successful molding, and the observation time points are 30 minutes, 60 minutes and 90 minutes. Why do you choose these time points? What is the basis?

4.In the control Group, lactic acid levels showed a significant downward trend, how to explain the reason? Based on these results, how to prove the effectiveness of the drug intervention in the experimental group?

5.The key indicator DµHbO2 is used in the chart(Fig 1 and 2), but not explain it in the method, what is the meaning of "D" here?

6.The lactate level decreased in the Control group and the three experimental groups, but the inconsistent performance of the microcirculation DµHbO2 level, especially the continuous decrease in the control group, which seems to be contradictory to our common sense. Why?

7.The dosage of PPAR-α antagonist GW6471 (1mg/kg/d) was chosen based on literature. However, this study did not show the inhibitory effect of this dose on PPAR-α under this experimental condition, which needs to be confirmed by supplementary experiments.

         Other tips:

        1. Charts need to be further standardized;

        2. The language needs to be more concise.

        3. Paragraphs on the limitations of this study were lacking.

Reviewer 3 Report

Dear authors,

The paper is well written a interesting. There are some minor errors in the text as well as in the figurs and I have made some suggestions for graphical representation in the attached pdf-file.

Statistics:

A power analysis could give an assessment of how small an effect could have been detected at maximum with this approach.

The questions supposed to be answered by this study were

·         - further elucidate mechanisms of statin action

·        - evaluate the effects of statins on the intestinal and hepatic microcirculation and mitochondrial function

·       - evaluate the possible role of PPAR-α-blockade in these interactions

Have these questions and aims been addressed?

"An effect of pravastatin on mitochondrial function is lacking."

To address this question, the setup with PPAR-α antagonist would not have been necessary. Instead of combining the factors PRAV and GW into a new one with 4 factor levels, it could possibly be worth to examine PRAV and GW as two different factors, resulting in a “3-Way-ANOVA” with timepoint, PRAV (y/n) and GW (y/n) as factors or two-way ANOVA with just time point and PRAV as factors? Especially, when it is assumed, that pravastatin is not acting through the respective receptor.

The “elucidation of mechanism of statin action” in this study is limited to the exclusion of the involvement of the PPAR-α receptor, although ways for alternative pathways and possible topics for future studies have been discussed.

Round 2

Reviewer 1 Report

The authors provide a thorough point by point rebuttal and carefully introduced the indicated modifications. The added limitations paragraph is especially wellcome.

The presentation of background, experiments, and data now much better allows the reader to judge the work. Thank you for the opportunity to review the manuscript.